# Age, Disease Severity and Ethnicity Influence Humoral Responses in a Multi-Ethnic COVID-19 Cohort

**DOI:** 10.3390/v13050786

**Published:** 2021-04-28

**Authors:** Muneerah Smith, Houari B. Abdesselem, Michelle Mullins, Ti-Myen Tan, Andrew J. M. Nel, Maryam A. Y. Al-Nesf, Ilham Bensmail, Nour K. Majbour, Nishant N. Vaikath, Adviti Naik, Khalid Ouararhni, Vidya Mohamed-Ali, Mohammed Al-Maadheed, Darien T. Schell, Seanantha S. Baros-Steyl, Nur D. Anuar, Nur H. Ismail, Priscilla E. Morris, Raja N. R. Mamat, Nurul S. M. Rosli, Arif Anwar, Kavithambigai Ellan, Rozainanee M. Zain, Wendy A. Burgers, Elizabeth S. Mayne, Omar M. A. El-Agnaf, Jonathan M. Blackburn

**Affiliations:** 1Department of Integrative Biomedical Sciences, Faculty of Health Sciences, University of Cape Town, Cape Town 7925, South Africa; muneerah.smith@uct.ac.za (M.S.); MLLMIC052@myuct.ac.za (M.M.); andrew.nel@uct.ac.za (A.J.M.N.); SCHDAR006@myuct.ac.za (D.T.S.); BRSSEA001@myuct.ac.za (S.S.B.-S.); 2Neurological Disorders Research Center, Qatar Biomedical Research Institute (QBRI), Hamad Bin Khalifa University, Qatar. Foundation, Doha P.O. Box 34110, Qatar; habdesselem@hbku.edu.qa (H.B.A.); ibensmail@hbku.edu.qa (I.B.); nmajbour@hbku.edu.qa (N.K.M.); nvaikath@hbku.edu.qa (N.N.V.); anaikjana@hbku.edu.qa (A.N.); kouararhni@hbku.edu.qa (K.O.); 3Proteomics Core Facility, Qatar Biomedical Research Institute (QBRI), Hamad Bin Khalifa University, Qatar Foundation, Doha P.O. Box 34110, Qatar; 4Sengenics Corporation, Level M, Plaza Zurich, Damansara Heights, Kuala Lumpur 50490, Malaysia; t.myen@sengenics.com (T.-M.T.); n.diana@sengenics.com (N.D.A.); science@sengenics.com (N.H.I.); Priscilla@sengenics.com (P.E.M.); r.shirin@sengenics.com (R.N.R.M.); shiela@sengenics.com (N.S.M.R.); a.anwar@sengenics.com (A.A.); 5Hamad General Hospital, Hamad Medical Corporation, Doha P.O. Box 3050, Qatar; Mariamali@hamad.qa; 6Anti-Doping Laboratory Qatar, Sports City Road, Aspire Zone, Doha P.O. Box 27775, Qatar; VALI@adlqatar.qa (V.M.-A.); Mohammed.AlMaadheed@adlqatar.qa (M.A.-M.); 7Virology Lab, Level 2, Block C7, Infectious Disease Research Centre, Institute for Medical Research, Setia Alam, Selangor 40170, Malaysia; kavithambigai@moh.gov.my (K.E.); rozainanee@moh.gov.my (R.M.Z.); 8Division of Medical Virology, Department of Pathology, University of Cape Town, Cape Town 7925, South Africa; wendy.burgers@uct.ac.za; 9Wellcome Centre for Infectious Diseases Research in Africa, University of Cape Town, Cape Town 7925, South Africa; 10Institute of Infectious Disease and Molecular Medicine, Faculty of Health Sciences, University of Cape Town, Cape Town 7925, South Africa; 11Department of Immunology, National Health Laboratory Service (NHLS) and University of the Witwatersrand, Charlotte Maxeke Johannesburg Academic Hospital, Johannesburg 2196, South Africa; elizabeth.mayne@nhls.ac.za

**Keywords:** immunoassay, SARS-CoV-2 nucleocapsid protein, epitope coverage, quantitative antibody binding, protein microarray, SARS-CoV-2 antibodies, humoral response

## Abstract

The COVID-19 pandemic has affected all individuals across the globe in some way. Despite large numbers of reported seroprevalence studies, there remains a limited understanding of how the magnitude and epitope utilization of the humoral immune response to SARS-CoV-2 viral anti-gens varies within populations following natural infection. Here, we designed a quantitative, multi-epitope protein microarray comprising various nucleocapsid protein structural motifs, including two structural domains and three intrinsically disordered regions. Quantitative data from the microarray provided complete differentiation between cases and pre-pandemic controls (100% sensitivity and specificity) in a case-control cohort (*n* = 100). We then assessed the influence of disease severity, age, and ethnicity on the strength and breadth of the humoral response in a multi-ethnic cohort (*n* = 138). As expected, patients with severe disease showed significantly higher antibody titers and interestingly also had significantly broader epitope coverage. A significant increase in antibody titer and epitope coverage was observed with increasing age, in both mild and severe disease, which is promising for vaccine efficacy in older individuals. Additionally, we observed significant differences in the breadth and strength of the humoral immune response in relation to ethnicity, which may reflect differences in genetic and lifestyle factors. Furthermore, our data enabled localization of the immuno-dominant epitope to the C-terminal structural domain of the viral nucleocapsid protein in two independent cohorts. Overall, we have designed, validated, and tested an advanced serological assay that enables accurate quantitation of the humoral response post natural infection and that has revealed unexpected differences in the magnitude and epitope utilization within a population.

## 1. Introduction

On the 30 January 2020, a public health emergency was declared by the World Health Organization (WHO) following extensive laboratory tests that led to the identification of a novel coronavirus, SARS-CoV-2, as the causative agent of pneumonia in Wuhan, China [1]. The virus can be spread from person-to-person via direct transmission of respiratory droplets or indirectly via contact with contaminated surfaces [2]. A global pandemic was declared in March 2020, leading to extreme measures to control the spread of coronavirus disease 2019 (COVID-19) [3], which in turn has had a negative effect on global economies, medical infrastructures, and mental health [4]. This has increased the need to understand the kinetics of the immune response to COVID-19. As of 12 March 2021, the coronavirus has spread to 221 countries and territories, affecting 119,165,187 people globally, and has been the cause of approximately 2,642,905 deaths [5].

Certain comorbidities have been associated with more severe COVID-19 symptoms and worse disease prognosis; therefore, understanding the underlying mechanisms for disease progression, including innate and adaptive immune responses, is of utmost importance to protect vulnerable individuals [6,7]. Furthermore, both differences in gender and ethnicity may influence disease susceptibility and mortality [8]. Classically, antigen-specific T-cells are considered the first line of adaptive responses to a new viral infection and act to limit disease severity and control disease progression, with antigen-specific CD8^+^ T-cells able to target and kill virally infected host cells; direct T-cell killing of viral particles is however less common. By contrast, the proliferation of antigen-specific B-cells takes longer, since it requires help from cognate CD4^+^ T-cells, but results ultimately in the secretion of high-affinity antigen-specific antibodies that can directly opsonize viral particles in peripheral fluids and mucosal tissues, thereby targeting the virus for neutralization and/or eradication, as well as providing the basis for mucosal immunity against subsequent reinfection. B- and T-cell responses thus work in parallel and are likely equally important in primary SARS-CoV-2 infections. Interestingly, recent data from the UK COVIDsortium suggest that while most COVID-19 cases develop either neutralizing antibody or T-cell responses, the correlation between the magnitude of these responses is discordant [9]. This suggests that a more detailed understanding of both B- and T-cell responses in COVID-19 disease, as well as in subsequent immunity against re-infection by SARS-CoV-2, is still required.

In general, antigen-specific antibodies are expected to vary in titer between virally infected individuals and also to vary in target epitope and functionality—including neutralization activity (by blockade of viral-host receptor interactions), directing phagocytosis or complement-dependent killing, or agglutination. Following the COVID-19 outbreak, many antibody tests have been developed to determine the extent of current and previous SARS-CoV-2 virus infections in a given population. However, most of these antibody tests are qualitative or semi-quantitative mono-epitope tests and are unable to localize antibody binding or characterize the breadth of epitope coverage in individual patients. Given the current global interest in the age-dependence and durability of humoral responses to natural infection and to vaccination, there therefore remains a need for new, advanced serology assay platforms that can assist in quantifying the complexity of the antibody responses to COVID-19 disease.

Screening for immunoreactivity utilizing a high-throughput antigen microarray in principle enables the simultaneous assay of multiple discrete, folded domains and epitopes of a given antigen, thus potentially allowing identification of antibody correlates of on-going protection and of development of durable immunity against subsequent SARS-CoV-2 infection. Furthermore, using pre-pandemic and known negative samples, it is possible to identify sources of cross-reactivity, which can be utilized to re-engineer functional epitopes to decrease the rate of false positives; however, this risks decreasing the sensitivity by the removal of true target epitopes. Recent studies utilizing various protein array platforms have reported high specificity and sensitivity [10,11,12]; however, these previous platforms lack the ability to quantitate differential antibody epitope utilization—including both linear and discontinuous epitopes—across cohorts of convalescent COVID-19 patients.

In addition, due to the high sequence similarity between SARS-CoV-1 and SARS-CoV-2 [13], there is a potential for antibody cross-reactivity between SARS-CoV-1 antibodies and SARS-CoV-2 antigens in regions where the original SARS outbreak was prevalent. However, a previous study reported that SARS-CoV-1 specific antibodies were undetectable in 91% of samples tested six years following infection [14]. Furthermore, there were a total of only 8096 SARS-CoV-1 cases worldwide, and SARS-CoV-1 has not circulated in the human population for over 17 years [15]; therefore, the chances of false positives in serological assays due to cross-reactivity are very low. In contrast, the seroprevalence of antibodies against naturally circulating human coronaviruses (hCoVs) is ubiquitous in most individuals [16], making the possible immune cross-reactivity between the four common hCoVs (229E, NL63, OC43, and HKU1), SARS-CoV-1, MERS, and SARS-CoV-2 an important factor in the design of immunoassays.

Here, we have designed and validated a novel, quantitative, sensitive, and specific SARS-CoV-2 multi-epitope fluorescent immunoassay, based on the nucleocapsid protein. The array is based on the use of the biotin carboxyl carrier protein (BCCP), which acts as a marker for the correct folding of proteins, since only correctly folded proteins will be biotinylated. Therefore, it is possible to control the immobilization of antigens onto a streptavidin coated surface in an oriented manner [17]. Different prototype array designs, using various engineered SARS-CoV-2 nucleocapsid protein structural motifs, were tested on a cross-sectional convalescent COVID-19 cohort and pre-pandemic controls to determine cross-reactivity. The specificity and sensitivity of the final array design were validated in an independent cohort. We then used this SARS-CoV-2 antigen microarray platform to explore the relationship between clinical data—age, disease severity, and ethnicity—and quantitative, epitope-specific antibody titers in a cohort of COVID-19 patients drawn from a migrant worker population in a single geographic region.

## 2. Materials and Methods

### 2.1. Study Design

Three different COVID-19 cohorts were used to develop, validate, and utilise the immunoassay.

#### 2.1.1. Cohort 1

Serum or plasma were prepared from blood samples collected from a cross-sectional cohort of 106 convalescent COVID-19 patients, recruited from Gauteng and Western Cape, South Africa, and stored at –80 °C until further analysis. The clinical characteristics of this cohort are summarized in Table 1. These patients were originally tested for SARS-CoV-2 using reverse transcriptase polymerase chain reaction (RT-PCR), using upper respiratory tract samples (nose or throat). These serum/plasma samples were used to design and develop the prototype array platform. Ethical approvals for these studies were obtained from the Human Research Ethics Committees of the University of Witwatersrand (M200468) and the University of Cape Town (UCT; HREC 210/2020). All patients provided written, informed consent. The plasma of 58 pre-pandemic colorectal cancer (CRC) patients and 10 healthy volunteers were used as additional controls for developing the array platform (UCT ethics approval HREC 269/2011).

#### 2.1.2. Cohort 2

The validation study was performed using sera collected from fifty randomly selected, hospitalized, PCR-positive COVID-19 patients with severe disease as part of the standard of care at Hospital Sungai Buloh, Selangor, Malaysia. The clinical characteristics of the patients in the cohort are summarized in Table 1. Fifty pre-pandemic HIV positive serum samples were used as true negative controls. In this cohort, no additional clinical annotations were provided.

#### 2.1.3. Cohort 3

Hospitalized COVID-19 positive patients (*n* = 100) admitted to Hamad Medical Corporation hospitals in Doha, Qatar, with confirmed positive RT-PCR results (sputum and throat swab) for the SARS-CoV-2 virus were randomly selected and enrolled for this study. The demographics of this cohort were therefore expected to be representative of COVID-19 cases in Qatar and included individuals from various ethnic groups (Middle Eastern (Qatari), Middle Eastern (non-Qatari), South Asian, and other). Peripheral blood was collected within five to seven days of admission and processed into plasma and serum, and then stored at −80 °C until further analysis. Patients were classified as having either mild/moderate disease (*n* = 50) or severe disease (admitted to intensive care unit; *n* = 50). Four patients were deceased from the severe group. Blood samples from age, gender, and ethnicity matched healthy volunteers (*n* = 38) with no prior COVID-19 infection history and with normal oxygen saturation and vital signs were recruited by the Anti-Doping Laboratory Qatar (ADL-Q) for blood collection. Individuals with medical history or with cognitive disability were excluded. The clinical characteristics of COVID-19 and healthy participants are summarized in Table 1.

All participants (patients and controls) provided written informed consent prior to enrolment in the study. Ethical approval for these studies was obtained from the Hamad Medical Corporation Institutional Review Board Research Ethics Committee (reference MRC-05-003).

### 2.2. Selection, Cloning, and Expression of SARS-CoV-2 Antigens

#### 2.2.1. Antigen Selection for Immunoassay Platform

Full-length SARS-CoV-2 nucleocapsid protein (UniProt accession number P0DTC9), as well as the core structural domains of the N protein (annotated as N-core) (44–362 aa), the N- terminal domain (NTD) (43–179 aa), the C-terminal domain (CTD) (246–363 aa), and 17 tiling peptides consisting of predicted B-cell epitopes in the intrinsically disordered regions (IDRs; including peptides spanning residues 395–412, 211–228, and 367–389) were selected for inclusion on the prototype array design.

#### 2.2.2. Gene Synthesis and Cloning

The full-length SARS-CoV-2 nucleocapsid (N) gene was synthesized (GeneArt, Regensburg, Germany) and cloned into a proprietary *Escherichia coli/*
*Spodoptera*
*frugiperda* transfer vector, pPRO8, such that the construct encoded the full-length N protein as an in-frame fusion to a C-terminal Biotin Carboxyl Carrier Protein (BCCP) and c-Myc tag. pPRO8 is a derivative of pTriEx1.1 (Sigma, St Louis, MO, USA) and encodes the *E. coli* BCCP domain (amino acids 74–156 of the *E. coli accB* gene) downstream of a viral polyhedrin promoter and cloning sites; flanking this *polh*-BCCP expression cassette are the baculoviral 603 gene and the 1629 genes to enable subsequent homologous recombination of the construct into a replication-deficient baculoviral genome [17].

N-core, NTD, and CTD clones were constructed from the full-length N gene using the oligo pairs summarized in Appendix A. Amplicons were generated by polymerase chain reaction using Vent DNA polymerase (New England Biolabs, Ipswich, MA, USA), digested with *Spe*I and *Nco*I (New England Biolabs) restriction enzymes and ligated into the equivalent sites in pPRO8, using standard protocols. All generated clones thus encoded N-protein structural motifs as in-frame fusions to a C-terminal BCCP c-Myc tag. In addition, seventeen tiling peptides (‘IDRs 1 to 17’) were synthesized with an N-terminal biotin moiety (Synpeptide, Shanghai, China) (Appendix A).

#### 2.2.3. Expression of Nucleocapsid Proteins as Fusions to a BCCP Tag

Following co-transfection of *S. frugiperda* Sf9 cells with a relevant pPRO8-derived transfer vector plus a linearized, replication deficient bacmid vector (*Autographa californica* baculovirus vector pBAC10:KO_1629_ [17]), baculovirus was amplified and recombinant proteins were expressed in *S. frugiperda* superSf9–3 strain (Oxford Expression Technologies, Oxford, UK) using previously published protocols [17]. Clarified cell lysates were prepared in insect lysis buffer (25 mM Hepes, 50 mM KCL, 20% glycerol, 0.1% Triton × 100, 1 × Halt™ Protease Inhibitor Cocktail, EDTA-free (Thermo Scientific, Waltham, MA, USA), 0.25% sodium deoxycholate acid, 25 U/mL Pierce Universal nuclease (Thermo Scientific), pH 8). Expression yields and *in vivo* biotinylation of each antigen were assessed by Western blot using a streptavidin-HRP conjugate probe (GE Healthcare, Chicago, IL, USA) (Appendix A). Lysates were stored at −80 °C before array printing. Peptides were solubilized in the same buffer (without nuclease and protease inhibitor) at a final concentration of 0.1 mg/mL. Control antigens used in the microarray included 50 μg/mL of biotinylated human immunoglobulins G, A, and M (hIgG, hIgA, and hIgM, respectively; Rockland, Gilbertsville, PA, USA) and 132 μg/mL of biotinylated anti-human immunoglobulin G (anti-hIgG; Rockland) as well as in house derivatized NHS-ester-Cy3 (Thermo Scientific) biotinylated BSA (Cy3-BSA) at 40 μg/mL.

### 2.3. Fabrication of Prototype and Final Protein Microarray

Prototype microarrays were printed using a QArray2 printer (Molecular Devices, San Jose, CA, USA) using methods described previously [18] on proprietary streptavidin-coated hydrogel slides (7.5 × 2.5 cm; Sengenics Corporation, Singapore). Each antigen was printed in triplicate with a mean size of 450 µm per spot. Eight replica arrays were printed per slide. After printing, the slides were incubated in a blocking buffer (20% Glycerol, 25 mM HEPES buffer (pH 7.4), 50 mM KCl, 1% Triton X-100, 1 mM DTT and 50 μM Biotin) and stored at 4 °C until used.

The final array layout (Appendix A) was fabricated using piezo-electric printing technology (Biodot, Irvine, CA, USA) onto streptavidin-coated hydrogel slides. Each antigen was printed in triplicate in a 24-plex format (i.e., 24 replica arrays per slide) with a mean size of 125 μm per spot. Slides were blocked and stored at −20 °C in blocking buffer (25 mM HEPES, 50 mM KCl, 4 mM CaCl_2_, 20 mM MgCl_2_, 20% Glycerol, 0.2% Triton X-100, 2% BSA). Successful immobilization and in situ purification of biotinylated proteins from lysates were confirmed via an anti-c-Myc (Sigma) assay.

### 2.4. Serological Assays

#### Optimization of Serum Concentration and Determination of Linear Range

For serial dilution assays, the serum or plasma was diluted 1:50, 1:100, 1:200, or 1:400 before adding it to the slides and commencing with the hybridization assay, as described below. All prototype microarrays were developed measuring IgG responses using 20 μg/mL AlexaFluor (AF) 647-labeled anti-human IgG. Notably, we observe no significant difference in performance of our immunofluorescence assays with serum or plasma (data not shown) and consider the assay to be equally compatible with both.

Microarray slides were washed with PBST (PBS, 0.2% Tween-20, pH 7.4) at RT for 3 × 5 min with gentle agitation, then dried by centrifugation at 1200× *g* for 2 min. Individual arrays were isolated using ProPlate 24 plex multi-well chambers (GraceBio-Labs, Bend, OR, USA). Prior to assays, serum samples were incubated with 0.1% Triton X-100 for 1 h on ice to deactivate potential live virions, then diluted 1:50 in assay buffer (PBST, 0.1% BSA, 0.1% milk powder). Individual arrays were incubated with 50 μL diluted serum for 1 h at RT with gentle agitation, then briefly rinsed with PBST, after which the slides were removed from the gaskets, washed for 3 × 5 min in PBST and dried by centrifugation at 1200× *g* for 2 min. 

Arrays were then incubated with detection antibody (20 μg/mL Cy3-labeled anti-human IgG in assay buffer) for 30 min at RT with gentle agitation. The wells were briefly rinsed with PBST, after which the slides were removed from the gaskets and washed for 3 × 5 min in PBST with gentle agitation and dried by centrifugation at 1200× *g* for 2 min.

### 2.5. Bioinformatic Analysis

#### 2.5.1. Image Analysis: Raw Data Extraction

Slides were scanned at a fixed gain setting using either an InnoScan 710 (Innopsys, Carbonne, France) or G2505C (Agilent, Santa Clara, CA, USA) fluorescence microarray scanner, generating a 16-bit TIFF file. A visual quality control check was conducted, and any arrays showing spot merging or other artefacts were re-assayed.

A GAL (GenePix Array List) file containing information regarding the location and identity of all probed spots was used to aid with image analysis. Automatic extraction and quantification of each spot were performed using either Mapix software (Innopsys) or GenePix Pro 7 (Molecular Devices) software, yielding the median foreground and local background pixel intensities for each spot.

#### 2.5.2. Data Pre-Processing

The mean net fluorescence intensity of each spot was calculated as the difference between the raw mean intensity and its local background. Extrapolated data were filtered and normalized using an in-house developed software (CT100+ programme). CVs for biotinylated Cy3-BSA were routinely below 5%. Human IgG (detected by fluorescently labeled secondary antibody) and human anti-IgG (detected only when plasma or serum is added to the slide) were used as positive controls to assess image signal intensity. Thresholds for positive signals for each antigen were determined using the OptimalCutpoints package with an emphasis on maximizing specificity [19].

Reciprocal titers per-antigen were determined from measured net fluorescence intensity, based on the projected further dilution of the sample required to reach the limit of detection in the assay, according to the following equation:*Reciprocal Titer* = *(Net Intensity (RFU)* × *initial serum dilution/limit of detection (RFU))*(1)

Underlying assumptions include: linearity of antibody binding signal vs. serum dilution, as observed both in this work and previously on protein arrays with the same underlying architecture [20]; linearity of signal observed for the dilution series of biotinylated hIgG controls on protein arrays with the same underlying architecture, in accordance with ligand binding theory (data not shown); and an assumed limit of detection of 50 RFU (equating to the noise threshold of the surrounding background). A cumulative score was then calculated based on the sum of reciprocal titers for non-overlapping domains of the N antigens to determine the seropositivity of a given sample.

#### 2.5.3. Statistical Tests

Sensitivity, specificity, and confidence intervals estimate were estimated using previously reported methodologies [21]. Other statistical analyses and graphical representation were generated using the R programming language (v 4.0.2) and GraphPad Prism (v 9.0; GraphPad Software, San Diego, CA, USA). Pearson’s correlation was performed to establish correlations between cumulative titer and various variables. Either the Wilcoxon–Mann–Whitney test or a one-way ANOVA with Welches correction was applied to determine the statistical significance of the differences observed between multiple independent groups (HC, mild and severe or case vs. control).

## 3. Results

### 3.1. Developing a High-Sensitivity, High-Specificity SARS-CoV-2 Antigen Microarray

It has previously been estimated that roughly 90% of B-cell epitopes are discontinuous [22,23] and surface exposed, yet it is well known that antibodies have a propensity for binding non-specifically to normally buried hydrophobic surfaces that become exposed on unfolded proteins. In order to allow for antibody recognition of discontinuous as well as linear surface exposed epitope, while minimizing non-specific binding, we fused full-length and functional domains of the SARS-CoV-2 nucleocapsid protein to a C-terminal Biotin Carboxyl Carrier Protein (BCCP) tag and expressed the resultant fusion proteins in insect cells. BCCP is only biotinylated *in vivo* when correctly folded [24], and misfolded fusion proteins have been shown to result in misfolding of BCCP; thus, only correctly folded fusion proteins become biotinylated and bind to a streptavidin-coated surface [17].

#### 3.1.1. Selecting N-Protein Constructs for the Final Microarray Design

The IgG response to SARS-CoV-2 full-length N protein was compared between pre-pandemic healthy controls (HC) and convalescent COVID-19 patients (P) drawn from Cohort 1. A serial dilution (1:50, 1:100, 1:200, 1:400) of pooled samples from the 10 HC and 10 P samples was performed to assess overall signals (Appendix A). Although the signal is higher for the Ps than the HCs, high relative fluorescent units (RFU) signals were detected for both sample sets, which was confirmed for the individual HC and P samples as shown in Appendix A.

An additional three SARS-CoV-2 N-protein constructs were therefore cloned, expressed, and purified/immobilized on the microarray, corresponding to the core structural domains (‘N-core’; residues 44–362), as well as the isolated N-terminal domain (residues 43–179) and C-terminal domains (residues 246–363; Appendix A). Domain boundaries in the SARS-CoV-2 nucleocapsid protein were identified by ClustalW-based sequence alignment of the SARS-CoV-1 (UniProt ID: P59595) and SARS-CoV-2 (UniProt ID: P0DTC9) nucleocapsid protein sequences and comparison with published structures of the SARS-CoV-1 nucleocapsid protein (PDB IDs: NTD, 1SSK; CTD, 2CJR).

We determined the optimal serum concentration for antibody binding to these new antigens using a serum dilution series from 1:50 to 1:12800. Appendix A shows representative ligand (i.e., antibody) binding curves for two randomly selected samples from Cohort 1 (P189 and P192). For P189, the highest dilution that still gave signal above background for the three N-protein constructs was 1:6400 dilution, with signal beginning to saturate at 1:100 dilution (Appendix A). For P192, the highest dilution that still gave signal above background was 1:400, and signal was still in the linear range at 1:50 dilution (Appendix A). We used 1:50 serum dilution for all subsequent assays.

These additional protein constructs also allowed us to assess non-specific binding and epitope coverage. Here, selected plasma samples from eight colorectal cancer patients (Cohort 1) were used as disease controls (C) and compared to seven Ps (Appendix A). The RFU signals for Cs were similar, ranging from 786–3855 and 639–3376 RFU for the full-length N protein (no PLS) and truncated N protein, respectively. However, the RFU signal for Ps was higher for the truncated N protein (3615–36993 RFU) compared to the full-length N protein (3034–12405), suggesting that the truncated N protein could offer a similar level of specificity, but a higher level of sensitivity compared to the full-length N protein. The C- and N-terminal domains display lower levels of non-specific binding with RFU levels ranging from 154–1050 and 219–1684 RFU for the Cs, respectively. However, the RFU signal for the Ps also decreased, ranging from 1011–16845 and 560–5161 for the C- and N-terminal domains, respectively.

#### 3.1.2. Selecting Peptides from the N Protein for Microarray Fabrication

To further improve the sensitivity and specificity of the platform, and to determine epitope coverage, a microarray was fabricated with 17 biotinylated peptides (Appendix A) derived from the N protein, which were predicted B-cell epitopes [25]. The IgG response to these 17 peptides was initially assessed using 10 HCs and 15 Ps (Appendix A). Varying degrees of non-specific binding were observed for 14 of the peptides, whereas Peptides 2, 6, and 8 showed little or no non-specific binding for the HCs, and a linear response with serum dilution for Ps. Two peptides (Peptides 5 and 10, both of which are lysine- and arginine-rich and have strongly basic patches) were observed to bind non-specifically and with high titers to pre-pandemic disease control sera, as well as to anti-human IgG, anti-His, and anti-c-myc antibodies: these two peptides flank the core structural domains of the nucleocapsid protein and may thus explain the significant cross-reactivity of the full-length SARS-CoV-2 N protein observed here with pre-pandemic sera (Appendix A). Peptides 1, 3, and 16 showed some non-specific binding, but some Ps who were non-responsive to Peptides 2, 6, and 8 were found to be responsive to Peptides 1, 3, or 16. Thus, Peptides 1, 2, 3, 6, 8, and 16 were retained for further analysis.

To evaluate which predicted N-protein B-cell epitopes resulted in the highest frequencies of disease-specific antibody binding, samples from 91 Ps and 58 Cs were then assayed against Peptides 1, 2, 3, 6, 8, and 16 (Appendix A). Nine Ps (RFU range: 301–2885) and two Cs (RFU range: 843–2623) produced an IgG response to Peptide 1; 27 Ps (RFU range: 138–62833) and four Cs (RFU range: 165–18245) produced an IgG response to Peptide 2; 15 Ps (RFU range: 123–64465) and 11 Cs (RFU range: 122–7704) produced an IgG response to Peptide 3. Notably, the frequency of positive signals amongst the Ps to Peptides 1, 2, and 3 was relatively low, while the magnitude of the IgG signal from the majority of Ps to these peptides was also found to be low and in the same range as signal from the Cs, suggesting that these peptides were not suitable for further development. By contrast, 45 and 41 Ps, respectively, displayed a moderate to high IgG response to Peptides 6 and 8, while only four Cs displayed low IgG responses towards either (RFU range: 141–1012), indicating that these peptides individually should have a high specificity and a moderate sensitivity. Finally, although a median signal of ~2500 RFU was found with 12 Cs for peptide 16, 41 Ps produced signals > 5000 RFU, including a number of Ps that were not reactive to peptides 6 or 8, indicating that the signal from true positives was well above the non-specific binding threshold and that Peptide 16 thus provided useful incremental benefit over Peptides 6 and 8.

Serial dilution assays using samples P189 and P192 demonstrated linearity of IgG binding to Peptides 6, 8, and 16 in the range 1:400 to 1:50 (Appendix A). We therefore elected to retain Peptides 6, 8, and 16 in our design, as a means to maximize the sensitivity and specificity of the final microarray platform (Appendix A).

### 3.2. Technical Performance of the SARS-CoV-2 Antigen Microarray Platform in an Independent Validation Cohort

The IgG cumulative titer found for the 50 severe COVID-19 cases and 50 pre-pandemic controls in Cohort 2 was used to determine the specificity and sensitivity of the arrays. Patients were defined as seropositive towards COVID-19 when the reciprocal titer for one or more N antigens were elevated above a ‘Minimum Specificity = 1’ threshold determined using the OptimalCutpoints package, based on the pre-pandemic control data. All 50 hospitalized COVID-19 patients were found to be seropositive, and all 50 pre-pandemic controls were found to be seronegative on the microarray platform; thus, the performance accuracy of the array was calculated to be 100% (Table 2). Figure 1 further validates the accuracy of the array, as there is a significant elevation in antibody titers to all antigenic domains in all case samples compared to the pre-pandemic controls (Figure 1A).

### 3.3. Quantitative Analysis of an Independent, Multi-Ethnic Cohort Reveals Differences in Antibody Titers and Epitope Coverage Scores Associated with Age, Disease Severity, and Ethnicity

A significant increase in antibody titers was observed between individuals with mild or severe disease and healthy controls in a further independent, multi-ethnic cohort (Cohort 3) recruited in Qatar (Figure 1B). Notably, our data reveal that the dominant antigenic epitopes lie in the two structural domains (and particularly the C-terminal domain), rather than in the intrinsically disordered regions of the nucleocapsid protein for both mild and severe disease patients in Cohorts 2 and 3, as judged by both the magnitude (reciprocal titer) and frequency of antibody recognition of the different structural motifs on our platform (Figure 1).

In Cohort 3, the nominally healthy control samples were recruited during the pandemic, rather than pre-pandemic, and were individuals with no history of COVID-19 disease but who were not tested by PCR. Four of these 38 controls were called positive by our immunoassay (Table 3), initially suggesting a specificity of 89.5%. However, closer inspection revealed that three of these four seropositive samples show significant reciprocal titers against two or more non-overlapping epitopes on the N protein (Figure 1B and Figure 2), increasing the confidence in these controls being true positives. It therefore seems likely that these individuals in fact had prior asymptomatic SARS-CoV-2 infections, rather than representing false positive immunoassay results; the actual specificity of our immunoassay in Cohort 3 thus appears to be 97.4–100%.

The sensitivity of detection found amongst PCR positive cases with mild disease (58%) or severe disease (92%; Table 3) in Cohort 3 is at first sight in line with literature expectation. However, 85% of the samples (43/50 mild; 42/50 severe) were collected within the first 14 days post onset of symptoms, and all samples were collected within 5–7 days of hospital admission. A more detailed analysis of the time to seropositivity in Cohort 3 showed a sensitivity of 75% in the first seven days post symptom onset in patients who developed severe disease, increasing to 97% by day 14 (Appendix A), and a sensitivity of 56% by day 7 even in patients developing mild disease. This means that seropositivity was detected while those patients were likely still in the acute phase of infection, and we suggest that this relatively early, high sensitivity may reflect the low limits of detection achieved with our multi-epitope fluorescent immunoassay and draw attention to the high epitope coverage scores for the majority of both mild and severe seropositive patients as evidence for the basis of this technical performance (Figure 2). To further assess the performance of the assay in these five to seven day post positive PCR samples, the positive and negative predictive values were calculated and are given in Appendix A.

### 3.4. Elevated N-Specific Antibody Titers and Broader Epitope Coverage Observed in Patients with Severe Disease

To determine the breadth of the antibody response, the sum of the number of IgG positive epitopes was calculated for each sample and presented in Figure 2 as an Epitope Coverage (EPC) Score. Not only do patients with severe disease have significantly higher antibody titers than patients with mild disease (Figure 1B), they also respond to a broader range of epitopes (*p* = 0.00017; Figure 2). Furthermore, the majority of COVID-19 patients have a broader epitope coverage compared to healthy controls, and the differences in coverage are statistically significant for all comparisons (Figure 2B).

#### 3.4.1. Increasing Antibody Titers and Epitope Coverage with Increasing Age

In both Cohorts 2 and 3, a trend to increasing antibody titer was observed with increasing age, reaching statistical significance in Cohort 2 in the age 51–60 bracket (Figure 3 and Appendix A). A similar trend was observed for the breadth of the immune response, with patients over 40, over 50, and over 60 having increasingly elevated epitope coverage scores compared to patients under 40 in Cohort 2, reaching statistical significance in the age 51–60 (*p* = 0.042) and >60 (*p* = 0.029) brackets (Figure 4A). In Cohort 3, a similar trend of increasingly elevated epitope coverage scores up to age 60 was also observed in both mild and severe disease cases (Figure 4B), but the small number of patients over 60 (*n* = 6) precludes robust conclusions being drawn on whether there is a genuine decline in epitope coverage scores in the >60 bracket or not.

#### 3.4.2. The Influence of Ethnicity on N-Specific Antibody Titers and the Breadth of Epitope Coverage

The relationship between ethnicity, antibody titers, and epitope coverage was assessed, and the results are summarized in Figure 5. Of all ethnic groups assessed, the Middle Eastern ethnicity group, excluding Qatari, was the only group to display a significant increase in both antibody titers and epitope coverage in patients with severe disease in comparison to patients with mild disease (Figure 5).

Between patients with mild disease, South Asians have a significantly elevated antibody titer compared to the Middle Eastern ethnicity groups (Figure 5A). However, the same pattern is not observed between patients with mild disease for epitope coverage, and only the Qatari group has significantly narrower coverage in comparison to South Asians (Figure 5B). Both the Middle Eastern, excluding Qatari, and South Asian groups have significantly higher antibody titers compared to the Qatari group in patients with severe disease (Figure 5A). Interestingly, this trend is not reflected in epitope coverage, where the Middle Eastern group, excluding Qatari, has a significantly broader epitope coverage in comparison to the South Asian group (Figure 5B).

## 4. Discussion

In the current COVID-19 pandemic, there is increasing interest globally in obtaining a more detailed mechanistic understanding of the underlying immunology of COVID-19 disease at both the B- and T-cell level. A number of papers have described the existence and cross-reactivity of SARS-CoV-2 specific T-cell responses [26,27,28], as well as correlations with antibody responses [9]. Viral neutralization assays are now providing important new information on neutralizing antibody activity in individuals [29,30], but are typically lower throughput, so reported studies have been on smaller cohorts. Serology assays have thus to date been primarily used in seroprevalence studies to determine the extent of infection in populations, with the rapid serology tests that are typically used in such studies being characterized by qualitative data on single antigens and focusing on simple yes/no answers. Such tests are known to be strongly affected by the time delay between the acute phase of disease and measurement and are not well suited to answer more advanced serological questions such as how the magnitude and breadth of antibody responses varies with time through convalescence, with age or disease severity, or with ethnicity, in large cohorts.

However, with the global roll-out of the first COVID-19 vaccines now well underway, there is increasing interest in how age in particular influences the magnitude and durability of SARS-CoV-2 vaccine responses. In addition, the emergence of SARS-CoV-2 variants of concern, such as the B1.1.7 and B1.351 variants, which appear to allow for at least partial escape from pre-existing antibody responses, necessitates the development of new quantitative, high-throughput serological tools that are suitable to addressing questions about whether, for example, vaccination protects against infection in individuals, or whether (re)-infection can still occur, albeit with reduced disease severity. Quantitative, specific detection of the magnitude and breadth of humoral responses to SARS-CoV-2 antigens seems likely to shed new light on both of these questions.

SARS-CoV-2 encodes a number of major structural proteins that could in principle be used as the basis of next generation serological tests: the nucleocapsid (N), spike (S), envelope (E), and membrane (M) proteins. Recent literature using first generation serology tests suggests that anti-N IgG antibodies are more prevalent than anti-S IgG antibodies in COVID-19 cases and may therefore be better suited to population level studies [31]. However, despite the wealth of available COVID-19 literature, there are few data on anti-E or anti-M antibody responses, implying lesser applicability. Here, we have therefore chosen to focus on gaining a more detailed, quantitative understanding of how antibody responses to the nucleocapsid protein correlate with age, disease severity, and ethnicity.

To enable this, we have engineered a novel, quantitative multi-epitope SARS-CoV-2 protein microarray platform, removing specific nucleocapsid protein epitopes that flanked the structural domains and which were identified as binding strongly and non-specifically to multiple unrelated non-human monoclonal and polyclonal antibodies, yet preserving other more distal, highly discriminatory antibody epitopes in the intrinsically disordered regions. This design resulted in 100% sensitivity and specificity in discrimination of severe COVID-19 cases from pre-pandemic controls in an independent cohort derived from Malaysia. We then utilized this novel immunoassay platform in a cross-sectional multi-ethnic cohort derived from Qatar, consisting of confirmed COVID-19 cases with a gradation of disease severities as well as with a wide age distribution, and have made a number of unexpected observations about age and disease severity influences on the humoral response.

While there is a literature precedent for anti-SARS-CoV-2 antibody titers to increase with disease severity, as also found here in two independent cohorts, we also observed that the breadth of the antibody response—i.e., the number of discrete epitopes recognized per patient—also increased with disease severity (Figure 2), which makes intuitive sense in terms of the amplification of humoral response in individuals with high viral loads and more extensive, longer lasting infection and disease. Notably, the data also suggest that in both independent cohorts, the dominant antigenic epitopes lie in the C-terminal domain of the nucleocapsid protein, with that domain showing more frequent and higher antibody titers (Figure 1) compared to the N-terminal domain in both mild and severe cases. In contrast, antibody recognition of the intrinsically disordered regions appeared to have a lower frequency and lower titer—perhaps suggesting lower affinity of recognition of linear epitopes—supporting the hypothesis that discontinuous epitopes on the surface of the structural domains are the preferred antigenic epitopes on this viral protein and are key to the specificity of this platform.

Classically, older individuals are generally observed to be more susceptible to new infections, due to impairment of adaptive immune responses [32], including immune repertoire exhaustion [33], and deficiency in antigen-driven selection processes [34]. There is also evidence for quite different antibody responses to infection or vaccination in individuals over the age of 50, with differences reported in magnitude and affinity, as well as in antibody class/sub-class, somatic mutation intensity and efficiency, loss of B-cell diversity, and antibody poly-specificity [34,35,36]. There are thus significant concerns about how well SARS-CoV-2 vaccines will work in older, more vulnerable groups.

Here, disease susceptibility as a function of age in Cohort 3 mirrors expected trends, with adults in the age bracket of 20–40 years being under-represented and those over 50 years being significantly over-represented in the diseased cohort relative to the general population (*p* < 0.001; Table 4). However, unexpectedly, our data show that in Cohorts 2 and 3, both the magnitude and the breadth of anti-SARS-CoV-2 N-protein antibody response increases with age, relative to the under 40 age group, reaching statistical significance in the 51–60 age bracket (Figure 3 and Figure 4), although the small absolute sample numbers in the over 60 age bracket in both cohorts limited the interpretation of our data in that group. This observation might simply reflect increased disease severity in the older age groups, but a trend of increased epitope coverage in the age 51–60 bracket was observed in both mild and severe cases (Figure 4B), arguing that the ability to mount a strong and broad antibody response to SARS-CoV-2 is not compromised by age, at least in these two independent, ethnically diverse cohorts, which is encouraging for the effectiveness of vaccinations in elderly groups. At face value, there appears to be an age cut-off at 60, above which the epitope coverage is lower in Cohort 3, possibly due to impaired adaptive immune responses and/or immune exhaustion in this cohort. However, this is not observed in Cohort 2 and may simply be a function of low sample numbers in that age bracket in Cohort 3. Further research to understand whether the age-related changes observed here in antibody titer and breadth of epitope utilization manifest further in terms of affinity, class/sub-class, effector functions, durability, or poly-specificity of the resultant antibodies will be reported elsewhere.

The effects of ethnicity on SARS-CoV-2 infection and disease severity remain largely unknown [8]. Data reported by the Centre for Disease Control (CDC) suggest that COVID-19 disproportionally affects certain ethnicities [37]. However, due to other cofounding factors, such as socioeconomic factors and variable access to healthcare, it is challenging to determine whether there is an underlying mechanism to explain the observed disparities in the humoral response between different ethnic groups [8]. Here, amongst the PCR positive group from the Qatar cohort (Cohort 3), we observed significant differences in the magnitude and breadth of antibody responses between the different broad ethnicity groups. The Qatari population as a whole is comprised of ~10% Qataris and ~90% ethnically diverse migrant workers/expats (Table 4 and Appendix A), the latter of whom can be broadly grouped as being of South Asian, Middle Eastern, or ‘Other’ ethnicities. The entire Qatar population of ca. 2.8 m people live in a single highly localized geographic region and all have free access to health care, removing one of the confounders referred to above. Our initial expectation therefore was that we might observe a significant difference in antibody responses between individuals as a result of diverse genetic backgrounds or differing susceptibility to severe disease.

All ethnicities in Cohort 3 had higher cumulative reciprocal titers and high epitope coverage scores in severe compared to mild disease, as expected, which reached statistical significance in the non-Qatari Middle Eastern ethnicity group (*p* = 0.0045, reciprocal titers; *p* = 0.039, epitope coverage; Figure 5), but interestingly not in the Qatari group. Unexpectedly, we also observed a significant difference in reciprocal titers between the Middle Eastern (Qatari) and Middle Eastern (non-Qatari) severe disease groups (*p* = 0.0078; Figure 5). It seems reasonable to expect socioeconomic factors to play a role in the incidence of COVID-19 disease in this cohort; notably, females are significantly under-represented in the diseased cohort (*p* < 0.01; Table 4), while there is also significant under-representation of Middle Eastern (non-Qatari) and over-representation of Qatari COVID-19 cases relative to their proportions of the overall population (*p* < 0.05; Table 4), supporting this expectation. However, it is less immediately obvious whether or how socioeconomic factors might affect the humoral response following infection in severe disease cases. Given that the non-Qatari Middle Eastern group comprises nationals from Egypt, Sudan, Syria, Iran, and Yemen (Appendix A), it seems possible that genetic differences between the Qatari and non-Qatari Middle Eastern groups might underpin the apparently decreased magnitude of humoral responses following infection and increased risk of COVID-19 disease observed here for the Qatari group. While we did not have access to genome sequence data for this cohort to verify this, it is perhaps relevant that the Qatari population has been reported to have an elevated prevalence of common adult diseases [38], as well as of childhood autoimmune diseases such as type 1 diabetes [39], potentially suggestive of uncharacterized genetic factors that affect humoral immune responses through HLA allelic variation [40].

Amongst the migrant worker groups, we observed a significant difference between the non-Qatari Middle Eastern and South Asian groups, in terms of both reciprocal antibody titers (*p* = 0.0013 for mild disease) and epitope coverage scores (*p* = 0.0046 for severe disease), apparently at least qualitatively further supporting a role for genetic factors and warranting further investigation. Interestingly, the directionality of these comparisons differed between mild and severe disease: reciprocal titers and epitope coverage scores for the non-Qatari Middle Eastern mild disease group were lower than for the South Asian mild disease group, but were higher in the non-Qatari Middle Eastern severe disease group compared to the South Asian severe disease group. This may reflect a greater disease severity in the non-Qatari Middle Eastern group that was not captured by the clinical scores, but more likely again points to intrinsically different humoral responses to SARS-CoV-2 infection amongst the different ethnicity groups in Cohort 3. Further work to explore the underlying basis of these ethnicity-based differences in anti-SARS-CoV-2 humoral responses in a larger cohort, including through HLA allele sequencing, is thus now needed.

### Limitations and Further Work

Although this cross-sectional study is statistically powered and identified clear ethnicity-and age-associated differences in both antibody titers and epitope coverage, it is limited by the available cohort sizes, which meant that we were not able to divide the broad ethnic groupings more finely and that certain other ethnicities were essentially absent from the comparisons, while participants over 60 years were under-represented. Furthermore, Cohort 1 comprised convalescent COVID-19 cases with a significantly longer average delay between diagnosis and sample collection, a skewed demographic makeup that is not representative of the general population and with disparate access to healthcare, while Cohort 2 was designed for the case-control validation component of this study, so lacked the spectrum of disease as well as ethnicity data; collectively, these factors limited our ability to integrate results across the three cohorts.

In addition, the study is also limited by its exclusive focus on IgG antibody responses to the nucleocapsid protein. Future studies will expand our quantitative, epitope-resolved antibody assay platform to include the SARS-CoV-2 spike protein and clinically relevant variants thereof; we will also include detection of additional immunoglobulin classes (IgA and IgM) and sub-classes (IgG_1–4_; IgA_1–2_), as well as on-array Fc effector function and surrogate neutralization assays, in order to shed further light on the functional consequence of the differential antibody titers observed, particularly in older individuals. Longitudinal studies will enable assessment of the durability of the age-dependent phenomena reported here.

## Figures and Tables

**Figure 1 viruses-13-00786-f001:**
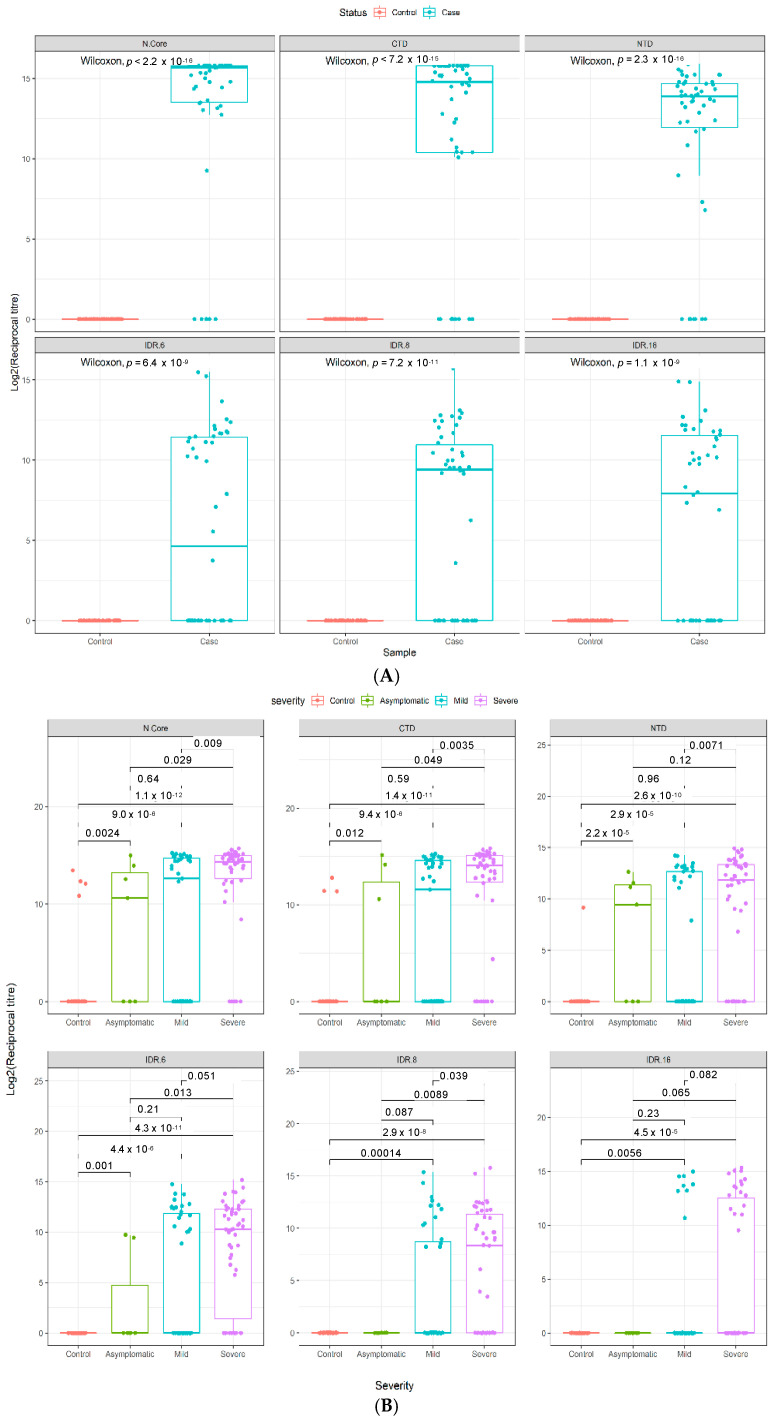
Epitope selectivity of IgG responses in two independent COVID-19 cohorts. Antibody reciprocal titers against different epitopes (*n* = 6) of the SARS-CoV-2 N protein in two separate COVID-19 case and control cohorts. (**A**) Validation cohort (*n* = 100), consisting of 50 hospitalized COVID-19 patients and 50 pre-pandemic controls (Cohort 2). (**B**) Multi-ethnic cohort (*n* = 138), consisting of 50 severe COVID-19 patients, 50 mild COVID-19 patients, and 38 healthy controls (Cohort 3). Boxes represent the 25th and 75th percentiles, and the midline represents the median and whiskers represent the 5th and 95th percentiles. *p*-values were determined using the Wilcoxon test (unpaired, two-tailed).

**Figure 2 viruses-13-00786-f002:**
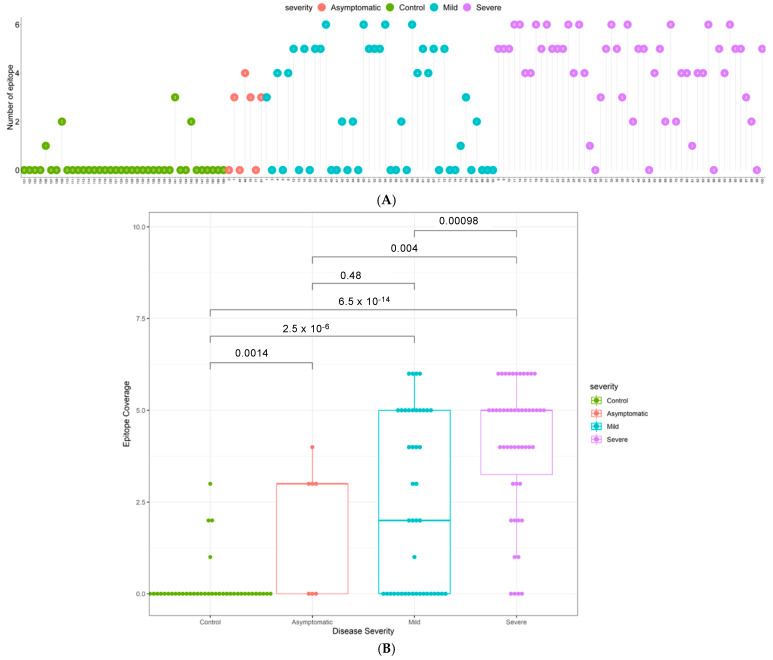
Epitope coverage in multi-ethnic Cohort 3. (**A**) Epitope coverage for each sample for controls, mild cases, and severe cases (*n* = 138). Numbers in dots represent the EPC score per participant. (**B**) Box plots displaying the epitope coverage for each disease class. *p*-values were determined using the Wilcoxon test (unpaired, two-tailed).

**Figure 3 viruses-13-00786-f003:**
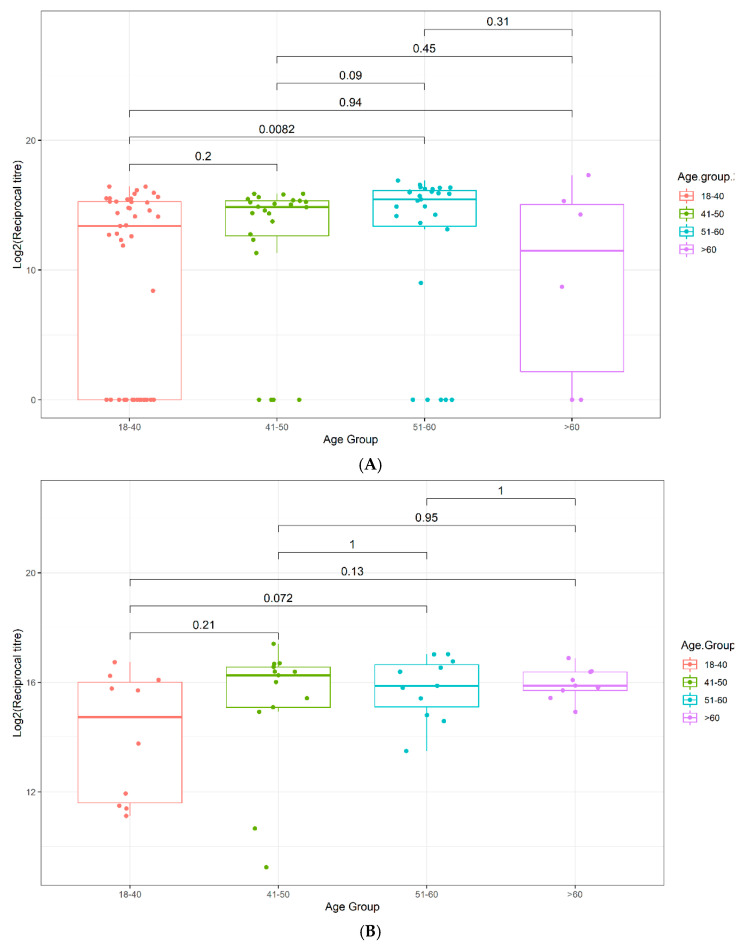
Box plots displaying the antibody reciprocal titers as a function of age. (**A**) Validation Cohort 2, (**B**) Multi-ethnic Cohort 3. *p*-values were determined using the Wilcoxon test (unpaired, two-tailed).

**Figure 4 viruses-13-00786-f004:**
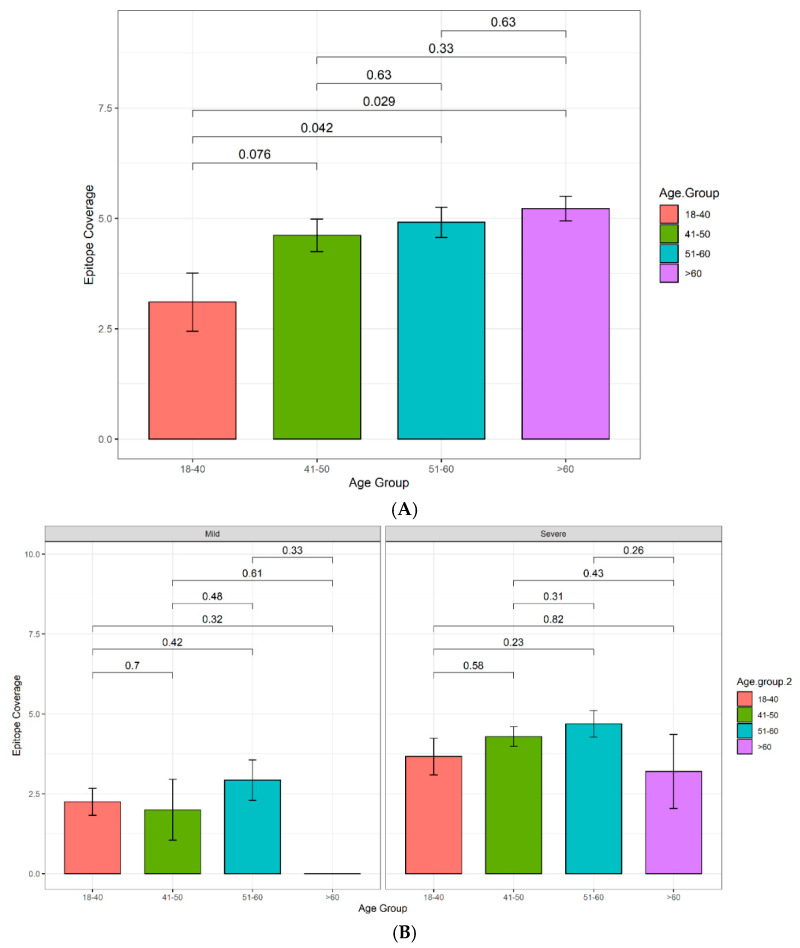
Histogram displaying the epitope coverage as a function of age. (**A**) Validation Cohort 2. Sample sizes: 18–40: *n* = 10, 41–50: *n* = 13, 51–60: *n* = 11, >60: *n* = 9. (**B**) Multi-ethnic Cohort 3; patients further categorized according to disease severity. Samples sizes: 18–40 mild: *n* = 28, severe: *n*= 15. 41–50 mild: *n* = 7, severe: *n*= 17. 51–60 mild: *n* = 14, severe: *n* = 13. >60 mild: *n* = 1, severe: *n* = 5.

**Figure 5 viruses-13-00786-f005:**
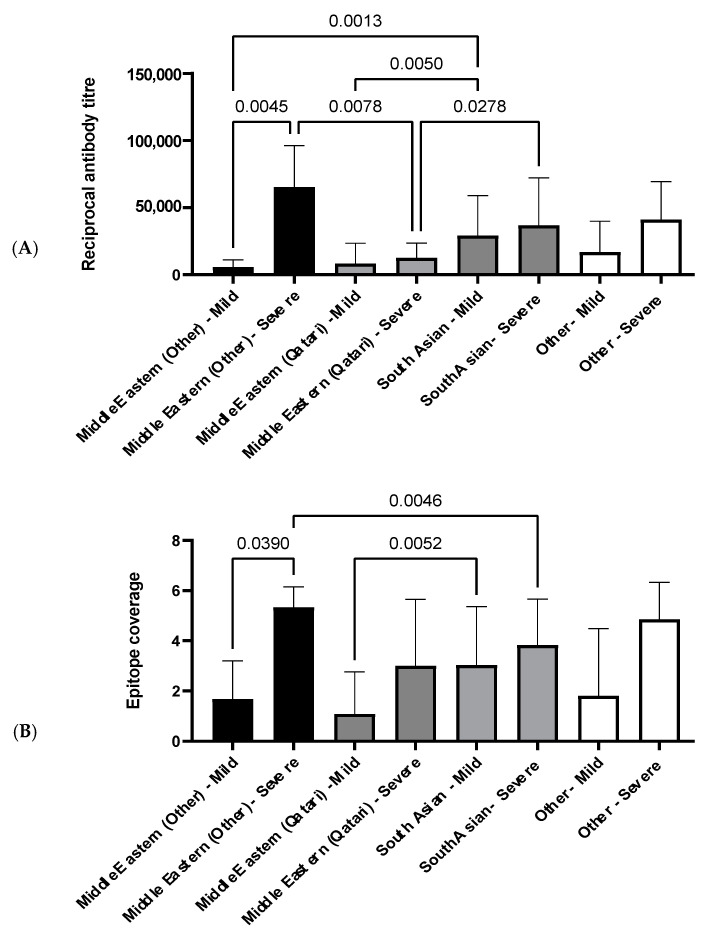
Histogram displaying relationship between antibody reciprocal titer and epitope coverage in Cohort 3. (**A**) Average antibody reciprocal titer for different ethnic groups and disease severities. (**B**) Average epitope coverage for different ethnic groups and disease severities. Pairwise comparisons were made using a one-way ANOVA, and *p*-values were calculated using Welch’s correction to compare the mean of each category with each other category. Sample sizes: Middle Eastern (other) mild: *n* = 2, severe: *n* = 6. Middle Eastern (Qatari) mild: *n* = 12, severe: *n* = 3. South Asian mild: *n* = 30, severe: *n* = 34. Other mild: *n* = 5, severe: *n* = 7.

**Table 1 viruses-13-00786-t001:** Clinical characteristic of COVID-19 patient cohorts.

Clinical Characteristics		Cohort 1	Cohort 2	Cohort 3
Total number of patients		174	100	138
Disease status	Pre-pandemic disease controls	68	50	0
	COVID-19 PCR − ve	23		
	COVID-19 PCR + ve	76	50	100
	No COVID-19 PCR test data	7		38
Disease Severity	Asymptomatic (PCR − ve)	4	0	0
	Symptomatic (PCR − ve)	19	0	0
	Asymptomatic (PCR + ve)	14	0	7
	Mild (PCR + ve)	24	0	43
	Severe (PCR + ve)	34	50	50
	Asymptomatic (no PCR test data)	7	0	38
	Not declared (PCR + ve)	4	0	0
Gender	Female	55 *	30 *	12
	Male	49 *	13 *	126
	Not declared	2 *	7 *	0
Age distribution	18–40	60 *	10 *	67
	41–60	38 *	24 *	65
	61–73	6 *	9 *	6
	Not declared	2 *	7 *	0
Ethnicity	African	9 *	0	
	Caucasian	72 *	0	0
	Colored	1 *	0	0
	Half-Japanese, half-Caucasian	1 *	0	0
	South Asian	9 *	100	94
	Middle East (Other)	0 *	0	10
	Middle East (Qatari)	0 *	0	18
	Other	0 *	0	15
	Not declared	14 *	0	1

* Convalescent PCR positive patients.

**Table 2 viruses-13-00786-t002:** Validation immunoassay data (Cohort 2). Confusion matrix showing the number of severe COVID-19 cases (*n* = 50) and pre-pandemic controls (*n* = 50) who gave a positive or negative assay result on the microarray platform, allowing calculation of clinical sensitivity and specificity.

Immunoassay Result	COVID-19 Status	
	Positive	Negative
Positive	50	0
Negative	0	50
	Sensitivity = 100%	Specificity = 100%

**Table 3 viruses-13-00786-t003:** Multi-ethnic cohort immunoassay data (Cohort 3).

Disease Severity	Immunoassay Result	RT-PCR Status		Sensitivity	Specificity
		Positive	Unknown		
**All samples** (Case, *n* = 100; Control, *n* = 38)	Positive	75/100	4/38	0.75	0.90
	Negative	25/100	34/38		
**Asymptomatic** (Case, *n* = 7; Control, *n* = 38)	Positive	4/7	4/38	0.57	0.90
	Negative	3/7	34/38		
**Mild** (Case, *n* = 43; Control, *n* = 38)	Positive	25/43	4/38	0.58	0.90
	Negative	18/43	34/38		
**Severe** (Case, *n* = 50, Control, *n* = 38)	Positive	46/50	4/38	0.92	0.90
	Negative	4/50	34/38		

**Table 4 viruses-13-00786-t004:** Summary of the demographics of Cohort 3 and the Qatari population. Percentage of each ethnic group in Cohort 3, compared to the percentage of each ethnicity found in the Qatari population. Ethnicities that did not fall under the three broader ethnic groups were excluded from this table (*n* = 5). Gender distribution in Cohort 3 compared to the gender distribution in the Qatari population. Age distribution in Cohort 3 compared to the age distribution in the Qatari population.

Characteristic	Number of Individuals in Cohort	Percentage of Cohort (%)	Percentage of Qatari Population (%)
**Ethnic Group**			
Middle Eastern (Other)	10	10	18.35
Middle Eastern (Qatari)	15	15	10.50
South Asian	70	70	64.32
**Gender**			
Male	91	91	72.90
Female	9	9	27.10
**Age Group**			
18–40	43	43	69.44
41–50	24	24	19.82
51–60	27	27	7.76
>60	6	6	2.99

## Data Availability

Data are contained within the article or Appendix A.

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
