# Peer review of "Age, Disease Severity and Ethnicity Influence Humoral Responses in a Multi-Ethnic COVID-19 Cohort"

_viruses, 2021, doi:10.3390/v13050786_

Round 1
Reviewer 1 Report
Very interesting manuscript concerning various factors of COVID-19 disease. The topic is very actual and for this reason, the work with real data should be considered as a priority. However, it needs to improve in order to be published:
- MAJOR POINT: the manuscript contains a lot of interesting data, however when I read it, it was like reading several different works. The data from each cohort are more-less represented individually and they need to be more connected together. I think it is fascinating, to combine and compare the risk of disease according to various race/ethnicity and age, however, a better connection is here necessary - figure 5 is not enough, also data Cohort 1 should be better explained. Also, in the discussion, several other works were published recently with similar topics (e.g. Kopel et al. 2020, Liang et al 2021) - authors should consider compiling and compare their results with others.
MINOR POINTS
- wrong nomenclature in the supplement (disease name is COVID-19, not Covid19 )
- The name of the article does not fit to the finding (represented in the current form) - something like 'multi-epitope SARS-CoV-2 protein microarray platform based on N-protein' will be more suitable
- several grammar issues were detected - please check the manuscript once again for minor grammar mistakes
- line 96 - explain 'hCov' and the text in brackets
- line 106 - too long and hard to understand
- The authors are not sufficiently explained, why they choose the N protein, why not e.g. S or E or M? Why they were focused on B-cells epitopes only? Maybe a few words in the intro/discussion about B-cells immunity in the COVID-19 will be beneficial, as most of the works consider T-cells immunity as crucial for the COVID-19 infections.
- Sample collection in the Method part need to improve: please include samples from Cohort 1 to table 1 (currently in supplement), and please include the Nr. of negative controls/individuals from each Cohort also to table 1 (no negative control cor cohort 3?).
- please add information about E. coli vector (company)
- the primers from 2.2.2 please serve in form of a table (in supplement)
- what kind of data is represented in the table under S2?
- Figure S3- interestingly, the control seems to interact also with the N proteins - some explanation why?
- Maybe I overlooked it, but - is there a list of which patient belong to which Cohort?
- Which software/platform was used to analyze the N protein (structural motif etc)?
- Figure 2 - it is necessary to increase the letter size in the legend. What is EPC? What means numbers in the dots? Please explain the analysis, - part 3.4
- Figure 3 is missing
- Why peptide 16 was chosen - as it seems to be very reactive in the control individual?
Author Response
Reviewer comment 1: The manuscript contains a lot of interesting data, however when I read it, it was like reading several different works. The data from each cohort are more-less represented individually and they need to be more connected together. I think it is fascinating, to combine and compare the risk of disease according to various race/ethnicity and age, however, a better connection is here necessary - figure 5 is not enough.
Author response: We appreciate the reviewer’s comment that the data from the three cohorts are largely represented individually and agree that it would be desirable to connect these as far as possible, in order to draw further inference regarding the risk of disease according to various race/ethnicity and age.
We have revised the relevant sections of the Discussion accordingly to draw together the data from Cohorts 2 & 3 a little more strongly – please see revised text on lines 583-590, 632-647 & 656-662.
We have also added text into the new Limitations and Further Work section, discussing the intrinsic differences between the cohorts and the available clinical/demographic data for each, which ultimately limits the ability to more deeply integrate the data from the three cohorts – please see revised text on lines 664-674.
In addition, we have added additional statistical analyses of the data from Cohort 3 into the manuscript to augment Figure 5, as requested – please see new Tables 4 and S5, which better explain Cohort 3 and the demographics of the cohort/population . We have now specifically assessed whether our data provides evidence of enhanced or reduced risk of COVID-19 disease amongst the different ethnicity groups, as well as assessing the gender- and age-associated risk of disease in this cohort - please see lines 583-586, 632-636 & new Table 4. We have also added in new text regarding the likely origins of the ethnicity-related differences observed in antibody titre and epitope utilisation - please see lines 637-647.
We therefore trust that these amendments and additional analyses will satisfy the reviewer’s request,
Reviewer comment 2: Data Cohort 1 should be better explained
Author response: Table 1 has been updated to include Cohort 1, as requested.
In addition, we have amended the text in the Results to clarify the results and the interpretation of the results from Cohort 1 – please see lines 332-339 & 353-389.
Reviewer comment 3: Also, in the discussion, several other works were published recently with similar topics (e.g. Kopel et al. 2020, Liang et al 2021) - authors should consider compiling and compare their results with others.
Author response: We have no incorporated and discussed Kopel et al. in the Introduction and more extensively in the Discussion, as requested. Please see lines 70-71 and 610-615 in the revised manuscript.
We were not able to find a relevant manuscript by Liang et al. (2021), but trust that the inclusion of Kopel et al satisfies the reviewer’s request.
Reviewer comment 4: Wrong nomenclature in the supplement (disease name is COVID-19, not Covid19)
Author response: This has been corrected and highlighted in the Supplementary file as requested.
Reviewer comment 5: The name of the article does not fit to the finding (represented in the current form) - something like 'multi-epitope SARS-CoV-2 protein microarray platform based on N-protein' will be more suitable
Author response: We respectfully disagree with the reviewer on this point; we have considered the comment but are satisfied that the existing title does accurately reflect the focus of the manuscript – which is on the new biological information gleaned through use of a next generation serology platform, rather than on the platform itself. We have therefore not changed the title.
Reviewer comment 6: Several grammar issues were detected - please check the manuscript once again for minor grammar mistakes.
Author response: We have checked and improved the grammar as needed throughout.
Reviewer comment 7. Line 96 - explain 'hCov' and the text in brackets.
Author response: We have explained hCoV as requested – please see line 119 in the revised manuscript.
Reviewer comment 8. Line 106 - too long and hard to understand.
Author response: We have amended that sentence to shorten and improve readability, as requested – please see lines 132-136 in the revised manuscript.
Reviewer comment 9. The authors are not sufficiently explained, why they choose the N protein, why not e.g. S or E or M? Why they were focused on B-cells epitopes only? Maybe a few words in the intro/discussion about B-cells immunity in the COVID-19 will be beneficial, as most of the works consider T-cells immunity as crucial for the COVID-19 infections.
Author response: We have amended that the text in the Introduction to discuss briefly the differing roles of B-cell and T-cell adaptive responses in viral infections and have provided a rationale for the present manuscript’s focus on antibody responses in COVID-19 disease, as requested – please see lines 71-90 in the revised manuscript.
In addition, we have amended the Discussion to explain why we chose to focus on antibody responses to the N protein rather than S, M or E proteins and have cited relevant literature to support this choice – please see lines 540-548 in the revised manuscript.
Reviewer comment 10. Sample collection in the Method part need to improve: please include samples from Cohort 1 to table 1 (currently in supplement), and please include the Nr. of negative controls.
Author response: We have amended that Table 1 in the main text to include Cohort 1 and all controls. We have also added in additional information on demographics of that cohort to amended Table 1 in the revised manuscript.
We have also amended the description of the sample collection as requested – please see line 142-144, 155-156 & 166-169 in the revised manuscript.
Reviewer comment 11. Please add information about E. coli vector (company)
Author response: We have added in a more detailed description of the E. coli and bacmid vectors and have provided references for each – please see lines 193-200 in the revised manuscript. We have also provide more detail and references for the protein expression protocols – please see lines 209-211.
Reviewer comment 12. The primers from 2.2.2 please serve in form of a table (in supplement)
Author response: We have moved the primers to Table S1 in Supplementary as requested.
Reviewer comment 13. What kind of data is represented in the table under S2?
Author response: We were not sure whether the reviewer meant Figure S2 or Table S2 so we have extended the legend to both Figure S2 and Table S2 to better describe the data contained therein. These changes are highlighted in the revised Supplementary file.
Reviewer comment 14. Figure S3- interestingly, the control seems to interact also with the N proteins - some explanation why?
Author response: We have included additional text in the Results to explain the origin of the cross-reactivity seen with the full-length N protein in Figure S3 – please see lines 359-365 in the revised manuscript.
The origin of this cross-reactivity was engineered out in the subsequent modified constructs that were used in the final design, as already described in the manuscript.
Reviewer comment 15. Maybe I overlooked it, but - is there a list of which patient belong to which Cohort?
Author response: We have checked and confirmed that the composition of each cohort is adequately described. We only refer to specific samples for Cohort 1, not Cohorts 2 or 3, and we are satisfied that there is no ambiguity there.
Reviewer comment 16. Which software/platform was used to analyze the N protein (structural motif etc)?
Author response: We have amended the Results to clarify the software and source files used to define domain boundaries - please see lines 327-331 in the revised manuscript.
Reviewer comment 17. Figure 2 - it is necessary to increase the letter size in the legend. What is EPC? What means numbers in the dots? Please explain the analysis.
Author response: We have increased the Legend size of Figure 2 and have explained EPC – please see lines 452-454. We have also explained the numbers in the dots in a revised legend for Figure 2: please see lines 475-476 in the revised manuscript.
Reviewer comment 18. Figure 3 is missing
Author response: We are not sure why Figure 3 went missing before since it was definitely in the Manuscript and Figures files we uploaded. We have re-inserted Figure 3.
Reviewer comment 19. Why peptide 16 was chosen - as it seems to be very reactive in the control individual?
Author response: We have amended the text in the Results section to make it more clear why we chose to retain Peptide 16 in the final design – please see lines 353-389 and specifically lines 381-389 in the revised manuscript.
Reviewer 2 Report
Dear Authors
The manuscript is well written and bring very strong information Age, disease severity and ethnicity.
In the introduction try to add the information from khurshid et al 2020, this study is the source for the spread of SARS-CoV-2 through saliva and support your introduction.
- Khurshid, Z., Asiri, F. Y. I., & Al Wadaani, H. (2020). Human saliva: non-invasive fluid for detecting novel coronavirus (2019-nCoV). International journal of environmental research and public health, 17(7), 2225.
Also, rephrase the line 291-310. Long statements and a lot of information try to make it easy for readers.
Figure-1: picture resolution is low. Try to fix it.
Heading 3.3.2. is not bold or italic.
Authors have to write the limitation and pitfalls of this study as well as the future direction of this research.
Author Response
Reviewer comment 1: In the introduction try to add the information from khurshid et al 2020, this study is the source for the spread of SARS-CoV-2 through saliva and support your introduction.
Author response: We have included reference to Khushid et al on lines 59-60 of the revised manuscript, as requested.
Reviewer comment 2: Rephrase the line 291-310. Long statements and a lot of information try to make it easy for readers.
Author response: We have re-phrased these two paragraphs as requested, shortening the statements and clarifying the information contained therein. Please see lines 324-339 of the revised manuscript.
Reviewer comment 3: Figure-1: picture resolution is low. Try to fix it.
Author response: We have improved the resolution of Figure 1 as requested.
Reviewer comment 4: Heading 3.3.2. is not bold or italic
Author response: This heading style has been corrected.
Reviewer comment 5: Authors have to write the limitation and pitfalls of this study as well as the future direction of this research
Author response: Although there already was a brief discussion of the limitation of the study in the original manuscript, we have now drawn this out in to its own section and have also expanded it to make more explicit additional limitations which we had assumed were implicit before. We have also included a few sentences on the future direction of this work, as requested. Please see lines 663-683 of the revised manuscript.